# Development of a Multiple Temperature Sensors Device for the Characterization, Control and Monitoring of Microbiological Incubators

**DOI:** 10.3390/s24237671

**Published:** 2024-11-30

**Authors:** Carolina Salinas Domján, Mauro A. Valente, Marcelo R. Romero

**Affiliations:** 1Instituto de Física Enrique Gaviola (IFEG), CONICET-UNC, Córdoba X5000HUA, Argentina; mauro.valente@unc.edu.ar; 2Laboratorio de Investigación e Instrumentación en Física Aplicada a la Medicina e Imágenes de Rayos X, LIIFAMIRx, FAMAF-UNC, Córdoba X5000HUA, Argentina; marcelo.ricardo.romero@unc.edu.ar; 3Centro de Excelencia en Física e Ingeniería en Salud & Departamento de Ciencias Físicas, Universidad de la Frontera, Temuco Casilla 54-D 4811230, Chile; 4Facultad de Ciencias Químicas, UNC, Córdoba X5000HUA, Argentina; 5Instituto de Investigación y Desarrollo en Ingeniería de Procesos y Química Aplicada, IPQA-CONICET, Córdoba X5016GCA, Argentina

**Keywords:** data logger, temperature sensors, ARDUINO, microbiological incubator, calibration

## Abstract

An increasing number of projects require the precise knowledge and control of thermal conditions within the study system and their temporal evolution. This is particularly critical for equipment such as laboratory ovens and microbiological incubators, which are essential in various fields of chemistry and microbiology areas. These devices allow and facilitate the execution of experimental work in controlled environments, leading to reproducible experiments. This work presents a methodology for assembling and calibrating a highly accessible and low-cost data logger equipped with multiple temperature sensors. The final calibrated dispositive is straightforward to construct and allows the simultaneous and independent temperature measurement from multiple positions within the same system, which is then applied to the study, characterization, control, and monitoring of the internal thermal behavior of a laboratory oven dedicated to microbiological agents’ cultivation. This approach ensures, through a robust methodology, a precise characterization by quantitative methods that allows objective decision making in the management and control of the temperature inside the system. Additionally, the device is suitable for extension and application in diverse research environments by modifying the sensor calibration to achieve a desired temperature range or number of measurement units, representing a potential work tool for laboratory systems.

## 1. Introduction

Nowadays, laboratory ovens have become an essential piece of equipment in areas such as chemistry, medicine, and microbiology, among others [1,2,3,4,5,6,7,8,9,10], since they facilitate the execution and reproducibility of experimental work. With ongoing advancements in science, numerous emerging projects now require precise temperature information within the studied system, since this magnitude can significantly impact various parameters [11,12,13] and, consequently, the outcomes of the research being conducted.

As shown in Figure 1, microbiological incubators are typically constructed from aluminum or stainless steel for optimal heat transmission and feature a metal door that provides thermal insulation across the front of the unit. The internal heat is generated by sets of electrical resistors located at the bottom, which transfer the thermal energy to the internal chamber. The resistor is a high-temperature region, while the chamber is a lower-temperature region [14]. The basic design consists of an insulated box with a heating system and adjustable thermostat. Usually, the most commonly used temperature is 37 °C, since most of the studied microorganisms develop well under that temperature [15], as it is the case with some bacteria and mammalian cells.

Generally, the internal temperature of such equipment is measured through a single integrated point temperature device regardless of whether the incubator corresponds to a recent model or not. Therefore, if any deficiency in the heat generation, or its control, occurs during equipment operation, this may go unnoticed and generate a lack of temperature homogeneity, producing high-impact consequences on the growth and development of the microorganisms of interest.

The internal temperatures of these microbiological incubators can also be measured using alternative measurement devices, such as data loggers, which record data over time or in relation to location by means of their own or externally connected instruments and sensors. However, these devices are usually small and have a single sensor or probe to measure temperature. Hence, the previously indicated problem remains unsolved.

It is within this context that the present work emerges with the objective of reporting a methodology for assembling and calibrating a data logger device that is simple to construct, low cost, and capable of simultaneously measuring temperature from different positions, facilitating a three-dimensional characterization of temperature in space and its evolution over time, allowing to know the changes produced during the operation of the microbiological incubator. Likewise, the proposed device is employed in the study, characterization, control and monitoring of the internal thermal behavior of a laboratory oven dedicated to cultivating microbiological agents, using a robust methodology that includes quantitative analysis. This methodology can be adapted for use in various systems, including ovens, by simply modifying the temperature sensor component to achieve the desired temperature range.

## 2. Materials and Methods

### 2.1. Device Elaboration

In the present work, the elaborated device is configured by using LM35 [16] analog temperature sensors connected to an ARDUINO UNO (Rev3 with Long Pins, Arduino SA, Switzerland) board.

The device to be developed is intended to be permanently operated according to typical microbiological incubators’ requirements, i.e., temperature ranging from 20 to 45 °C, as well as being low-cost and simple to elaborate, along with independent temperature measurement by each sensor with a response time of at least 1 s. The device is intended to be fully supplied with voltages suitable for the ARDUINO UNO board without additional adaptations, making it suitable for extension and application in various systems.

ARDUINO is an open-source electronics creation platform based on free hardware and software that allows the creation of different types of single-board microcomputers for multiple applications. The boards include all necessary components to connect peripherals to the inputs and outputs of a microcontroller, and they can be programmed on Windows, macOS, and Linux operating systems [17]. In addition, the temperature sensors attached to the board typically consist of a sensitive element, a protective cover, a filler, and a temperature conductor to transform temperature changes into electrical signals, which are processed by electronic equipment. The LM35 sensors provide a temperature-proportional voltage output in °C with a linear reading factor increasing the value at the rate of 10 mV per °C. Thus, the output is, for example, 0 mV at 0 °C, it is 250 mV at 25 °C and 400 mV at 40 °C. Temperature is assessed by sensors’ readouts due to the basic operation principle of diodes, as sketched in Figure 2. It is well known from semiconductor physics that the voltage across the diode increases at a known rate as the temperature increases, generating a signal directly proportional to the thermal variations [18]. Thus, the Shockley equation mathematically describes the relationship between the diode current (I) and the absolute temperature (T) according to the following expression:(1)$$I=I_{S}\left(e^{qV/kT}-1\right)$$
where $I_{s}$ is the reverse bias saturation current, *q* is the electron charge, *V* is the voltage across the diode and *k* is the Boltzmann’s constant [19].

It is worth mentioning that LM35 sensors have a measurement range of −55 °C to 150 °C with an accuracy of 0.5 °C at room temperature ($T_{amb}$) [16,20,21].

The board used in this study is an open-source microcontroller board based on the ATmega328P Microchip Technology (Atmel Corporation, San Jose, CA, USA) with an integrated circuit capable of recording instructions written with a variant of the C++ programming language [22,23].

In the first instance, the aim is to configure the system to measure temperature in at least four positions simultaneously. This number may vary depending on the positions required and the control board used given the available analog inputs. Figure 3 presents the configuration C proposed for the LM35 sensor (the schematics of configurations A and B can be found in Appendix A), while the three types of connections proposed between each LM35 sensor and the ARDUINO UNO board are illustrated in Figure 4.

Configuration A involves connecting each LM35 sensor connected to the ARDUINO UNO board, the $+V_{s}$ pin connected to the 5 V supply voltage, the pin GND to ground, and the data pin $V_{out}$ to the analog pin. In configuration B, a 47 nF multi-layer ceramic capacitor is added in parallel in the configuration with one end connected to the $+V_{s}$ pin and the other to the sensor’s GND pin. Finally, configuration C includes both the capacitor of configuration B and a 2 kΩ resistor connected in series to the $V_{out}$ pin of the sensor.

The connections of each sensor to the ARDUINO UNO board are consistent across configurations. Each sensor uses a 60 cm and 4 mm diameter *Audiopipe HWY 836 USA standard MC-1* shielded cable with two wires.

As depicted in Figure 5, the first wire connects the $+V_{s}$ pin of the sensor to the 5 V supply voltage of the board. The second wire connects the $V_{out}$ pin of the sensor to one of the analog inputs on the board. Finally, the shielding mesh is used as a third wire by connecting the GND pin of the sensor with the corresponding ARDUINO GND.

In Table 1, the list of necessary components to construct the device with configuration C, which includes the components of configurations A and B, is shown.

Once the device is assembled, it is connected via USB to a computer, to which the Arduino IDE 2.0.3 software was previously installed (downloadable on its website [24]). The corresponding code developed that allows the reading of temperatures and their recording in an external file is executed.

This code consists of three different sections: in the first one, the variables to be read and used are defined, in the second one, the *void setup* function is defined, where the commands that the program will execute during the system initialization are specified; finally, in the third section, the *void loop* function is written. The *void loop* function is the main function that contains the commands that will be executed while the board is enabled [24]. To read an analog sensor, *analogRead* can be used. It provides a value between 0 and 1023, from which a mathematical formula is obtained to calculate the temperature as a voltage function.

Figure 6 shows an example of the applicable code for one sensor. To read multiple sensors simultaneously, simply add the analogous lines of code per sensor. Once the code is completed, it is uploaded to the ARDUINO board. The temperature values measured by each configuration are recorded and compared, defining the one with the least variations and uncertainties during measurements in environments with constant temperature.

### 2.2. Device Calibration

With the selected configuration, all sensors are placed in the same space at $T_{amb}$, and measurements are made after 10 seconds (s) of the device stabilization. Temperature values per sensor are recorded every 1 s for 2 min. Subsequently, the $S_{j}$ values recorded by each *j* sensor are averaged ($P_{S_{j}}$) and compared with the average temperature value recorded from three mercury thermometers ($P_{T}$), which were previously calibrated and located in the same position as the sensors. The difference between these averages indicates the correction value ($C_{j}$) that should be applied to each $S_{j}$ to $T_{amb}$, as indicated by Equation (Equation 2).
(2)$$C_{j}=P_{T}-P_{S_{j}}$$

This measured value at ($T_{amb}$) corresponds to the first point in the table ($P_{S_{j}};C_{j}$), which was constructed alongside the measurements made in the system in Figure 7. A container of water is placed in a LabKlass 78HW-1 (Zenith Lab Inc., Jintan, China) magnetic stirrer and heated homogeneously with temperature values ranging from 32 to 45 °C recorded. Each sensor is placed in a second container to avoid contact with water and moisture. A fifth container, containing mercury thermometers, is also included in the system. Once the system reaches a first thermal equilibrium at 32 °C, the temperature measurements from the sensors and thermometers are recorded according to the methodology previously described.

Once the measurements are completed, the values ($P_{S_{j}};C_{j}$) are plotted. Based on the trend of the correction values, the correction equation $f_{j}\left(S_{j}\right)$ is formulated for each sensor. This correction equation is applied in the Arduino code for each corresponding sensor, as indicated by Equation (Equation 3), where $T_{j}$ represents the calibrated temperature per sensor.
(3)$$T_{j}\left(S_{j}\right)=S_{j}+f_{j}\left(S_{j}\right)$$

The suitability of calibration is measured using sensors and mercury thermometers in the same position at $T_{amb}$ and in a central position in the interior of a laboratory oven.

### 2.3. Stove Characterization

After developing and calibrating the data logger, the internal thermal behavior of the laboratory oven is characterized. The oven features a painted sheet metal exterior and has two stainless steel shelves. Its internal dimensions are 60 cm wide (X-axis), 40 cm deep (Y-axis) and 40 cm high (Z-axis), as shown in Figure 8. It is equipped with a W3230 thermostat [25,26] that controls the temperature. Its screen indicates both the current temperature and the configured temperature, which are measured by a sensor located at the base of the oven.

The temperature (T) stabilization time ($t_{s}$) is determined by taking measurements with and without the top shelf. The sensors are placed in the same position in the upper central area of the oven below the ventilation grille. The oven is set to 37 °C with a 2 °C hysteresis (minimum operating temperature of 35 °C).

Once the thermal stabilization is achieved, the evolution of the system is studied. Given that the data logger is equipped with 4 sensors, and the interior of the oven is divided into 5 blocks ($B_{j}$), each measuring 12 × 40 × 40 ${\mathrm{cm}}^{3}$. Each block is further subdivided into 4 rows and each row into 4 columns, as shown in Figure 9. This configuration creates four volumes (0 to 3) each measuring 12 × 10 × 10 ${\mathrm{cm}}^{3}$ by row with the sensors $S_{j}$ positioned at the center of these volumes. A total of 80 measurement positions is obtained, considering the three-dimensional layout.

Figure 10 represents the communication diagram for measurements of the whole system with the sensors in their respective positions according to the segmentation indicated in Figure 9.

### 2.4. Proposed Device Evaluation Against Thermocouple

After developing and calibrating the data logger and the stove characterization is completed, the operation of the elaborate device is compared with the operation of a previously calibrated thermocouple.

Figure 11 reports 10 positions where temperature measurements were made with a thermocouple and the proposed device. The central positions (P5 and P10) correspond to measurements with all 4 sensors and the thermocouple in the same position simultaneously, and the other positions correspond to only one sensor with the thermocouple; an example can be observed in Figure 12. Likewise, the measurements were carried out at different stages of the operation of the stove to corroborate the change in temperature and if both the sensors and the thermocouple register these variations.

Measurements were made in a lapse of two minutes for each position, with acquisition every second in the case of sensors, and every 20 s for the thermocouple temperature. Once registered, the measurements of each sensor and position are averaged and compared.

## 3. Results

### 3.1. Device Elaboration

Among the sensor connection configurations A (AC), B (BC) and C (CC), configuration AC is theoretically ruled out. Directly connecting the sensors to the board via wiring exposes them to ground currents, making them highly susceptible to externally induced signal noise. The wiring capacitance creates a bypass from ground to the sensor input. Additionally, performance can be negatively affected by environments where the wiring can act as a receiving antenna and the internal junctions as rectifiers [16]. Therefore, incorporating a capacitor alongside the temperature sensor is considered a minimum requirement in the circuit.

A representative section of the results from measurements for BC and CC, performed at $T_{amb}$ with data recorded every 1 s for 2 min, is shown in Table 2 (by configuration, the behavior of each sensor is analogous to that shown in the table).

The BC configuration exhibits unstable behavior with significant variation between measurements reaching up to 70.11 °C. In contrast, the CC configuration demonstrates remarkable stability, with a maximum variation of 0.20 °C, which falls within the 0.5 °C instrumental error margin, and an average recorded temperature of 23.38 °C. Moreover, it is important to note that when the sensors are placed near the surfaces of lower and higher temperatures, the BC configuration shows no changes, whereas CC records the corresponding decreases or increases in temperature; therefore, the CC configuration has successfully facilitated the construction of the data logger and the effective implementation of its code.

### 3.2. Device Calibration

From the table ($P_{S_{j}};C_{j}$) constructed with CC, the results reported in Figure 13 were obtained for each sensor along with their uncertainties calculated by error propagation. In all cases, it is evident that the correction values have an exponential tendency, leading to the proposal that the correction function is an exponential function of the following type:(4)$$F\left(x\right)=a\cdot e^{bx}\;;\;x\equiv S_{j}$$

For each sensor, the corresponding exponential fit was carried out (Figure 13), yielding the values of a and b, shown in Table 3, for each case, along with their uncertainties.

Based on these data, the function $f(S_{j})$ is constructed. The correction was applied to the $P_{S_{j}}$ values, and these were compared with the $P_{T}$ values, which are represented in Figure 14. It can be observed that once the correction has been applied, the linear behavior of the sensors more closely aligns with the linear behavior of the thermometers. The uncertainties corresponding to each corrected value were calculated, resulting in a maximum uncertainty of 0.7 °C.

Once the code has been corrected, it is measured every 1 s for 2 min at $T_{amb}$ with sensors and thermometers positioned in the same location. These values are averaged and compared. The average temperature recorded per sensor is indistinguishable from the average of the mercury thermometers, considering instrumental uncertainties. Consequently, the effectiveness of the calibration is confirmed, both in the initial system and within a laboratory oven, indicating, in turn, that the device is suitable for use in laboratory systems within a proven response range of 20 to 45 °C.

### 3.3. Oven Characterization

#### 3.3.1. Without Top Shelf

Positioning all sensors under the vent of the stove and without the top shelf, the T is recorded every 1 s, obtaining the results shown in Figure 15. Although a specific time ($t_{s}$) is not reached, a repetitive pattern in temperature behavior is observed.

Once the oven is turned on and set to 37 °C, T increases until the control sensor *W3230* reaches 37 °C (time A in Figure 15), although this does not correspond to the T at the sensors location. Subsequently, T decreases until the *W3230* sensor reaches 35 °C (time B in Figure 15), at which point the resistors resume heat transmission, reaching a new temperature peak in the upper central area.

This process continues until the maximum and minimum temperature recorded per cycle become indistinguishable from those of the following cycle (considering instrumental error). This temperature cycle repeats every 3000 s until the oven is turned off. Therefore, to characterize the internal behavior of the oven, a measurement duration of at least 5000 s (about 83 min) is recommended.

After positioning the sensors in their corresponding rows, T is recorded every second for each sensor ($S_{j}$) over a period of 5000 s from the oven’s ignition. Measurements begin from row 1 (R1) of block 1 (B1) and then proceed to rows R2 to R4, repeating the process for the other blocks in ascending order.

The acquired data are displayed three-dimensionally per second, as illustrated in Figure 16 for time 0 s.

From these data, a Newton interpolation of 1 degree is performed to approximate the values of g(X)=T in Equation (Equation 5), with *T* as the temperature in °C, for the unknown positions X=(x,y,z) for each measured t, and $g(X_{j})$ is the temperatures of the measured positions $X_{j}$. The temperature is evaluated between two points with known data that are close enough for the variation to be approximately linear.
(5)$$g\left(X|X_{1},X_{2}\right)=g\left(X_{1}\right)+\frac{g\left(X_{2}\right)-g\left(X_{1}\right)}{X_{2}-X_{1}}\cdot \left(X-X_{1}\right)$$

This approach allows for the visualization of the oven’s general behavior as seen in the examples for different t values in Figure 17.

Regarding the temporal evolution of the temperature per measured point, all cases produce curves analogous to those shown in Figure 15. After 3000 s and from the subsequent minimum, the behavior stabilizes, T varying between a fixed minimum and maximum.

Once the t from which the behavior becomes repetitive is reached, the averages of maximum temperature ($T_{max}$) in each position are observed, as shown in Table 4.

It is notable that $T_{max}$ per block increases from B1 to B5, suggesting that as the position approaches the right zone of the oven along the X axis, the temperature is higher, with a variation range from 34.8 to 36.3 °C (1.5 °C difference). As can be seen in the averages from Table 4, a difference of 1.3 °C between walls is obtained.

Along the Z axis, a temperature decrease is observed from the base to the ceiling in B1, and the center of B3 and B4, while it increases in B2, B5 and the areas near the bottom and front of the oven in B3 and B4. Taking the average of the maximum temperature differences per sensor between B1 and B5 across rows, the temperature differences increase from R1 to R4, indicating greater stability at the base of the oven than in the upper part near the ceiling.

In the lower half of the Y axis, there is a temperature decrease from front to back in B1 and B2, while in B4 and B5, it increases; the same behavior is maintained in the upper half except for B2, where the temperature increases rather than decreases.

These results suggest the presence of currents that produce temperature variations, which are noticeable in Figure 17. Consequently, it is possible to resize the work area to a cube of 30 × 30 × 30 ${\mathrm{cm}}^{3}$ concentric with the center of the oven. The temperature variation is significantly reduced to 0.4 °C, with an average of 35.1 °C, so it is recommended to work within said zone.

Another noteworthy aspect is that although maximum temperatures of 36.0 °C are reached, these are only exceeded at two positions, so the configuration objective of reaching an average of 37 °C inside the oven is not achieved. When observing the GAPs per row, it is notable that in most cases, these are around 2 °C, corresponding to the hysteresis configured in the oven, suggesting a correlation. Adjusting the configuration could achieve the desired 37 °C.

#### 3.3.2. Without Top Shelf

Recording the T data every 1 s, with all sensors in the same position in the presence of the upper shelf, the results shown in Figure 18 are obtained. A t of approximately 5800 s is identified, beyond which the behavior becomes repetitive. Accordingly, the duration of the measurements with the top shelf in place should be at least 7200 s (120 min).

With the resizing of the work area, the T data of the positions indicated in Figure 19 are recorded.

Linear interpolation is used to approximate the new values of g(X)=T. This allows to visualize the general temperature behavior in this case. Examples for different t values are shown in Figure 20.

The temporal evolution of temperature yields curves similar to those shown in Figure 18 for all points above the shelf. For the points below the shelf, although the behavior is similar, the difference between temperature maximums and minimums is significantly greater with higher maximums but shorter durations within the same time interval.

The longer times recorded with the top shelf are attributed to its construction from stainless steel with 18 holes of 1 cm diameter each. This causes higher T values below the shelf compared to above it, as seen in Figure 20.

Once the stabilization time ($t_{s}$) is reached, the average T achieved is recorded in the area of interest, as shown in Table 5.

The data presented reaffirm that the maximum average temperature ($T_{MAX}$) reached under the shelf is higher than the ($T_{MAX}$) in the upper zone. Furthermore, the observed gaps indicate that none of them approach the 2 °C of hysteresis, suggesting that adjusting the oven configuration based on its hysteresis is not suitable for this case. The average $T_{MAX}$ in the area of interest is 34.2 °C, so the configuration objective of 37 °C is not achieved in this case either.

### 3.4. Proposed Device Evaluation Against Thermocouple

After the temperature measurements of each selected position inside the stove are made, they are averaged correspondingly and compared the sensor temperature values to the thermocouple values, as displayed in Table 6.

The disadvantage in these cases is the time it takes to acquire all the positions. Given that to measure temperature at 4 different points with a thermocouple, the process has required 4 times the time that the elaborate datalogger uses to complete the same task, highlighting that the problem initially proposed can be easily solved at a low cost, with simple preparation and affordable and generally available components.

It should be noted that in all cases, the values recorded by the sensors are within the values recorded by the thermocouple, which presents an uncertainty of ±1 °C. Furthermore, if the values registered for P5 and P10 are observed, their differences are within the uncertainties of the device sensors after calibration.

## 4. Discussion

### 4.1. Device Elaboration and Calibration

Among the proposed configurations, the one incorporating a capacitor and a resistor connected to each sensor has proven to be the most appropriate, allowing the successful assembly of the temperature-measuring device. This configuration presents less susceptibility to ground currents and greater stability compared to the other configurations.

With the addition of the respective operating code and the calibration corrections, the final device represents a potential tool for diverse environments where the precise control of temperature-related variables in a studied system is necessary even when handling measurements from multiple positions simultaneously.

### 4.2. Stove Characterization

The successful characterization of the internal thermal behavior of the laboratory oven has been demonstrated with the proposed device even under different measurement conditions. However, as none of the configurations achieved the target oven temperature of 37 °C, two new configurations are proposed:Without top shelf: To maintain the maximum operating temperature at 37 °C with a 2 °C hysteresis, and adjust the controller’s sensor temperature readings as needed, different corrections were made until a correction of −1.4 °C achieved the desired objective. It is preferable to use a rack for supporting samples that allows greater circulation heat compared to the oven’s upper shelf.With top shelf: Similarly, to maintain the maximum operating temperature at 37 °C with a 2 °C hysteresis and correct the temperature record of the controller sensor, a correction of −2.0 °C is considered appropriate. However, given the temperature variations caused by the upper shelf, a maximum operating temperature configuration of 37.8 °C with a hysteresis of 2 °C and a correction of −1.4 °C is considered convenient, adapting the conditions of the case 1.

In both cases, the inner edges of the oven will reach temperatures higher than the desired setpoint even if within the appropriately resized work area. Decreasing the hysteresis temperature is not recommended because it shortens the lifespan of the relay at the base of the oven.

Each case was evaluated over time, showing that once the stabilization time has elapsed, the desired configuration is achieved and consistently maintained for up to 14 days after the oven is turned on.

This precise and quantitative characterization of the oven’s internal thermal behavior offers a robust method for making informed decisions on material handling in microbiological equipment. By providing specific knowledge of areas within the system that exhibit isotherms or temperature variations, as well as the extent of these variations, the method improves reproducibility in related studies.

The oven characterization method, as well as the use of the proposed system, are capable of being extended to other systems that require similar temperature control and characterization by simply adapting the temperature sensor component to achieve the desired temperature range.

## 5. Conclusions

This work presents a methodology for assembling and calibrating a low-cost data logger. It is designed to be highly accessible to researchers and laboratories with limited resources due to its affordability and the widespread availability of its components. Furthermore, the device is straightforward to construct and enables simultaneous temperature measurements at multiple locations within the same system, facilitating a three-dimensional characterization of temperature and its temporal evolution. This capability allows for the volumetric determination of temperature changes. Additionally, the device significantly reduces the measurement time, up to a quarter, compared to a single sensor system, which would require to determine temperature values from different positions. The proposed device was successfully built and calibrated, and it was able to operate according to typical microbiological incubators’ requirements. Each sensor independently measures temperature with a response time of a second (s). The device operates on voltages compatible with the ARDUINO UNO board without additional adaptations, making it versatile for application in various systems.

Additionally, its application in laboratory systems is demonstrated through its use in the study, characterization, control, and monitoring of the internal thermal behavior of a laboratory oven dedicated to cultivating microbiological agents, which is followed by a robust methodology that includes quantitative analysis. In the absence of the oven’s upper shelf, the stabilization time is 3000 s, while in the presence of the shelf, the time extends to 5800 s. The temperature inside the oven tends to increase along the X axis, and heat currents are suggested by the behaviors observed along the Z and Y axes, prompting a recommendation to resize the work area to a cube of 30 × 30 × 30 ${\mathrm{cm}}^{3}$ concentric with the center of the oven, where the temperature variation is significantly reduced to 0.4 °C. In both cases, with and without the upper shelf, an average maximum temperature of 37 °C can be achieved by setting the maximum operating temperature according to each case. Therefore, the proposed device and its methodology represent a potential work tool for laboratory systems with potential adaptation and extension to other research environments for more locations by simply modifying the temperature sensor number or calibration range to achieve the desired measurements.

## Figures and Tables

**Figure 1 sensors-24-07671-f001:**
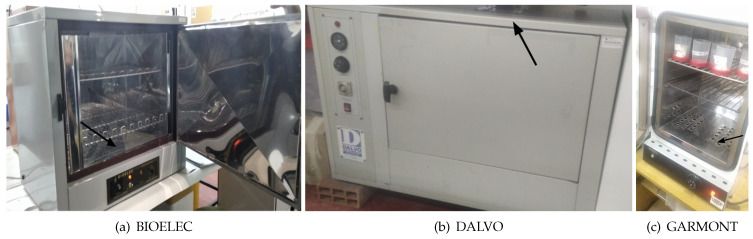
Examples of microbiological incubators and their temperature sensors positions indicated.

**Figure 2 sensors-24-07671-f002:**
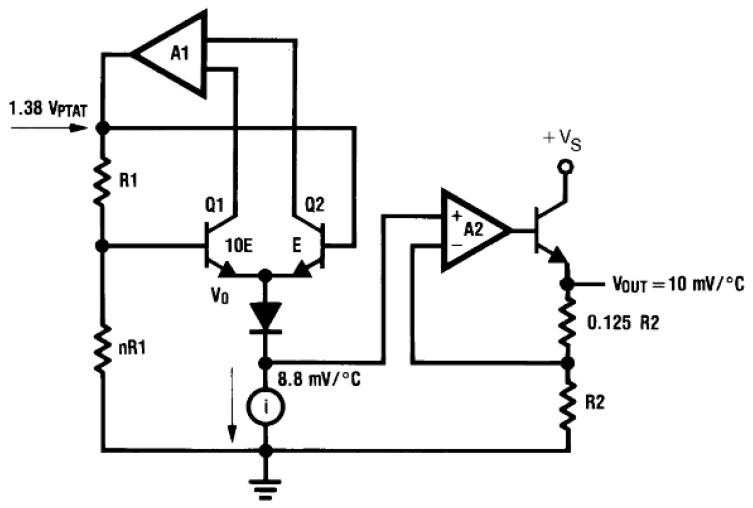
Internal schematic of LM35 temperature sensor according to bibliography [16,20].

**Figure 3 sensors-24-07671-f003:**
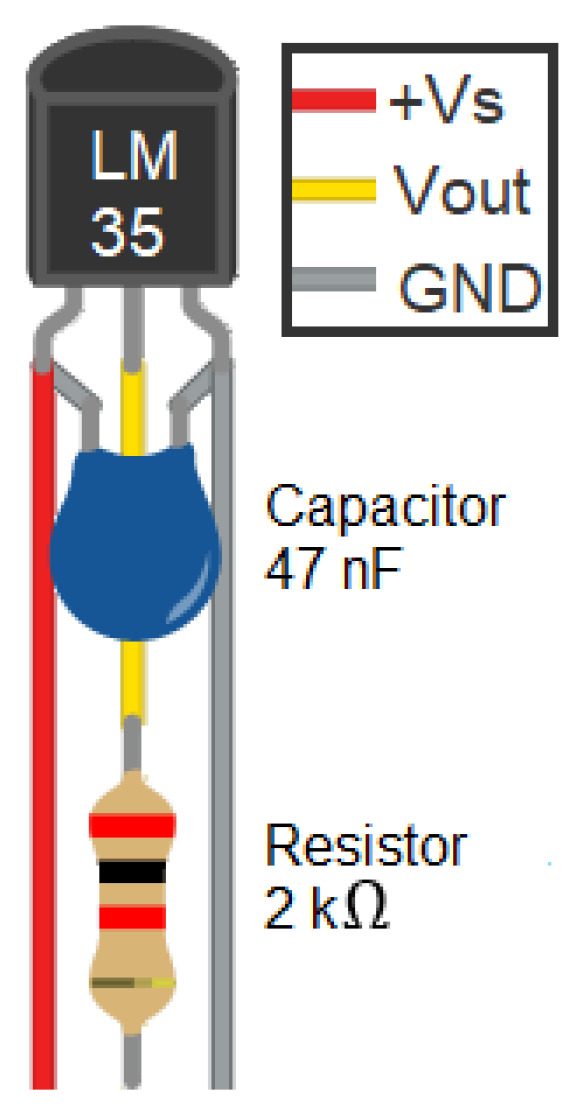
Configuration C proposed for the LM35 sensor.

**Figure 4 sensors-24-07671-f004:**
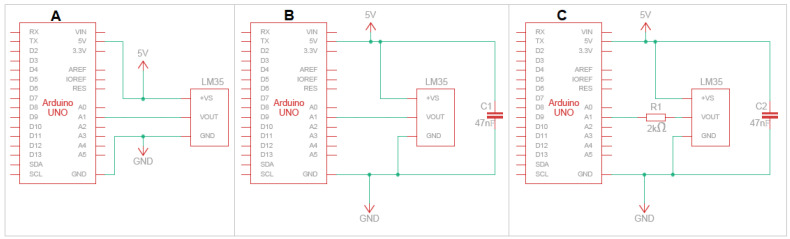
Electronic diagram of the proposed configurations for the LM35 sensors where A, B and C corresponds to configurations A, B and C, respectively.

**Figure 5 sensors-24-07671-f005:**
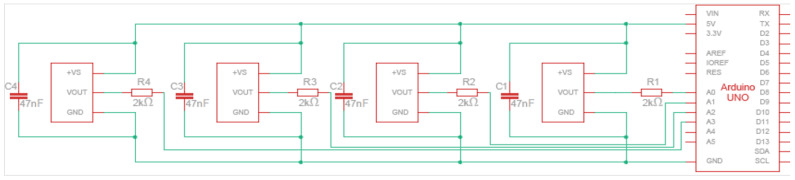
Circuit assembly example for configuration C.

**Figure 6 sensors-24-07671-f006:**
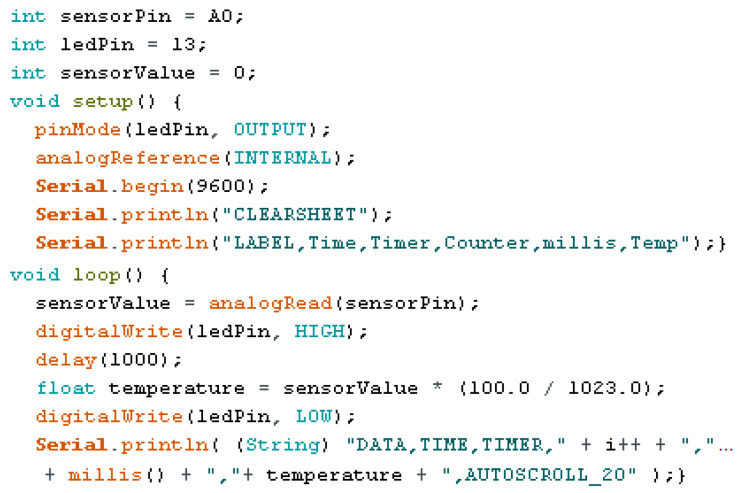
Example of code used for one sensor.

**Figure 7 sensors-24-07671-f007:**
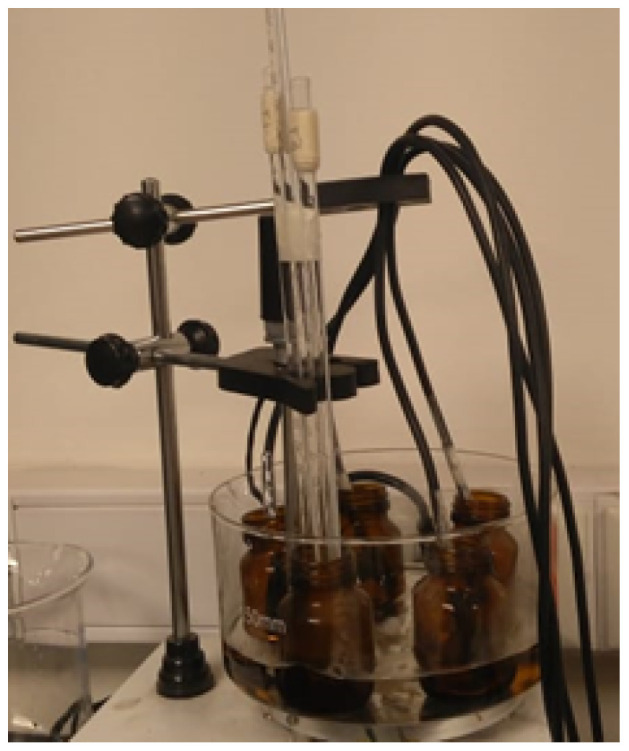
Experimental setup for calibration.

**Figure 8 sensors-24-07671-f008:**
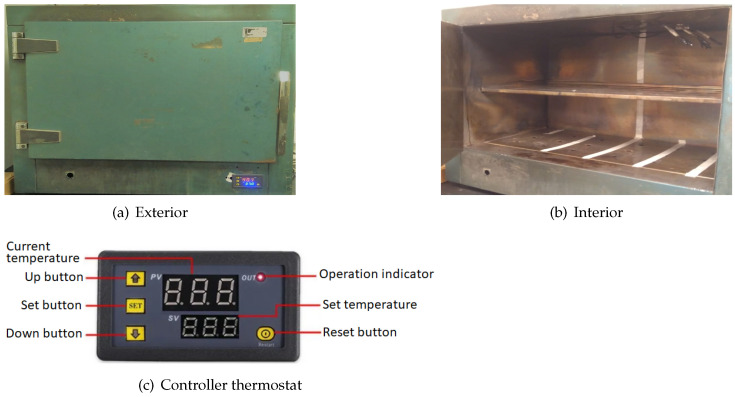
Bacteriological culture oven.

**Figure 9 sensors-24-07671-f009:**
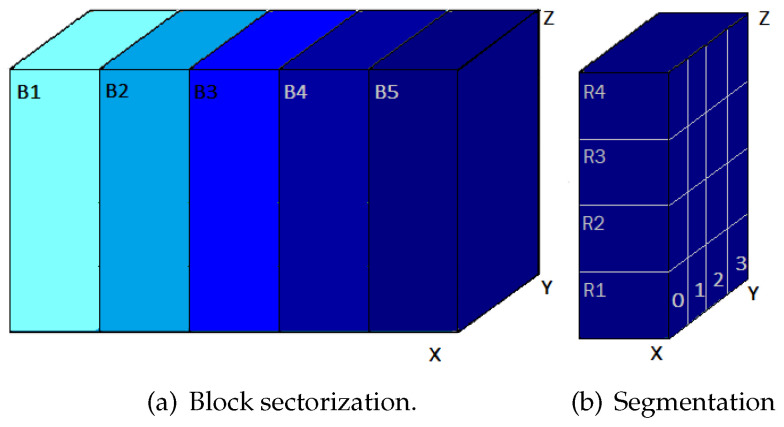
Stove division diagram.

**Figure 10 sensors-24-07671-f010:**
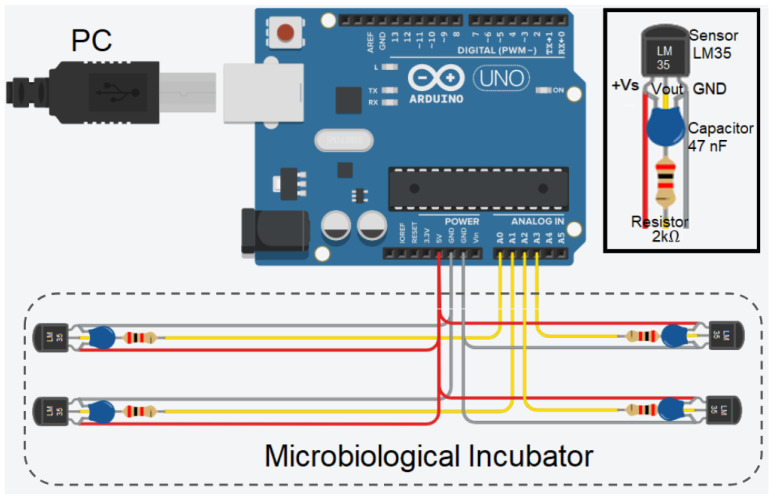
Diagram of whole system connections.

**Figure 11 sensors-24-07671-f011:**
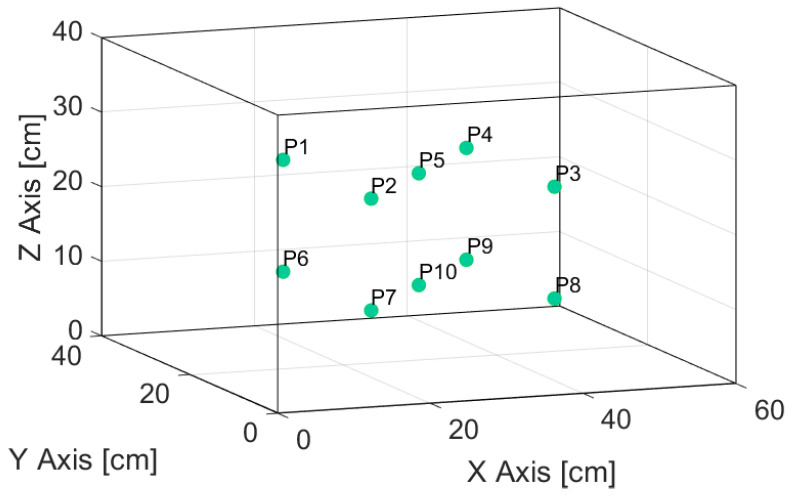
Sensors and thermocouple temperature measurement inside the oven.

**Figure 12 sensors-24-07671-f012:**
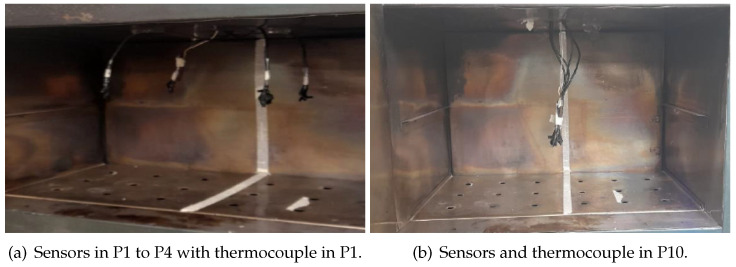
Example of positions inside the stove.

**Figure 13 sensors-24-07671-f013:**
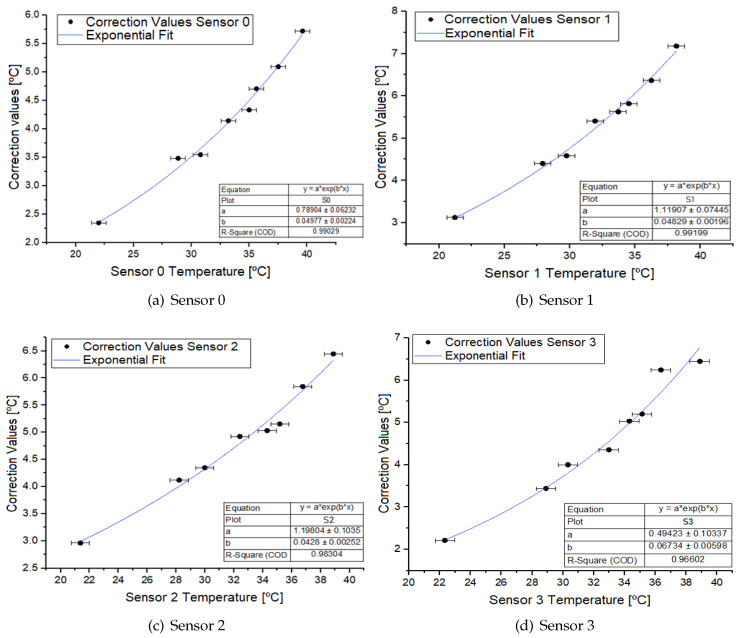
Correction values and their correspondent fitting curve by sensor.

**Figure 14 sensors-24-07671-f014:**
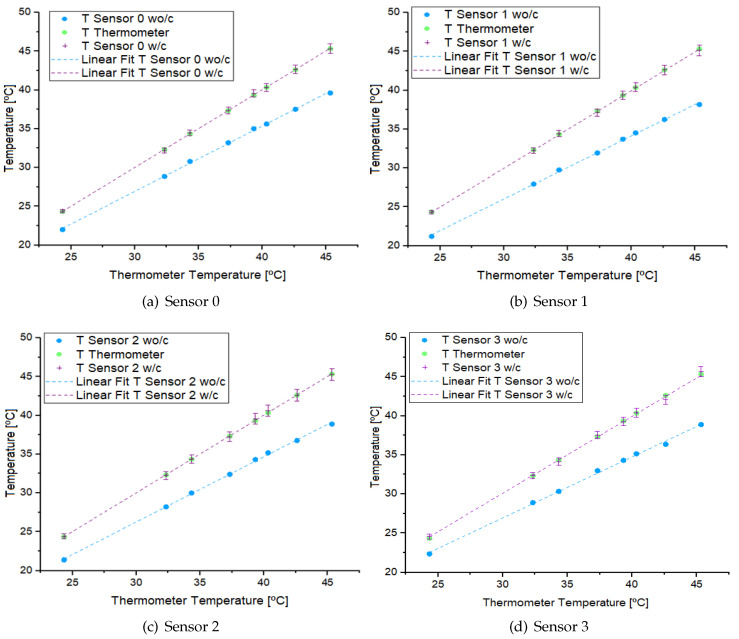
Behavior of sensors, with (w/c) and without (wo/c) correction, and thermometers.

**Figure 15 sensors-24-07671-f015:**
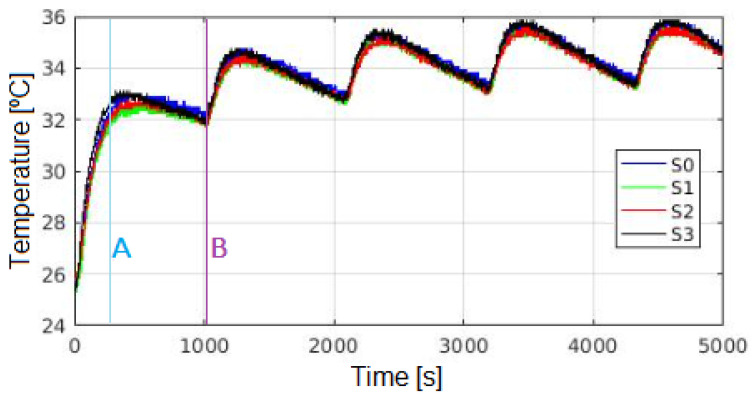
Values per sensor, without top shelf, including stabilization time.

**Figure 16 sensors-24-07671-f016:**
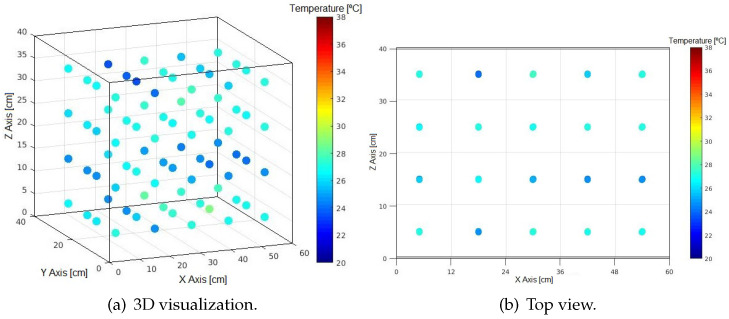
Measuring positions without shelf.

**Figure 17 sensors-24-07671-f017:**
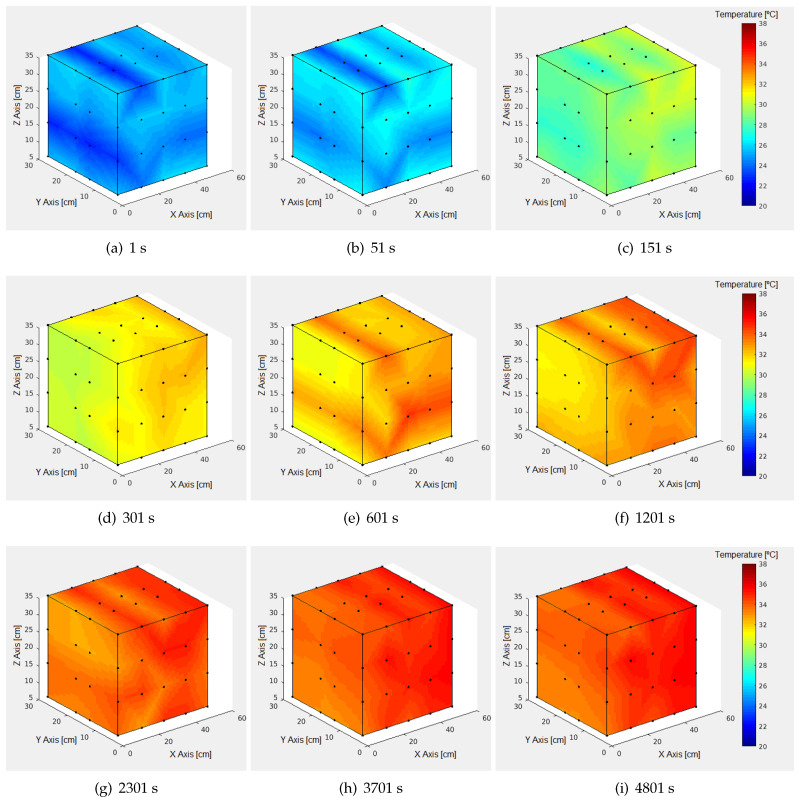
Internal thermal behavior of laboratory oven by time in seconds (s).

**Figure 18 sensors-24-07671-f018:**
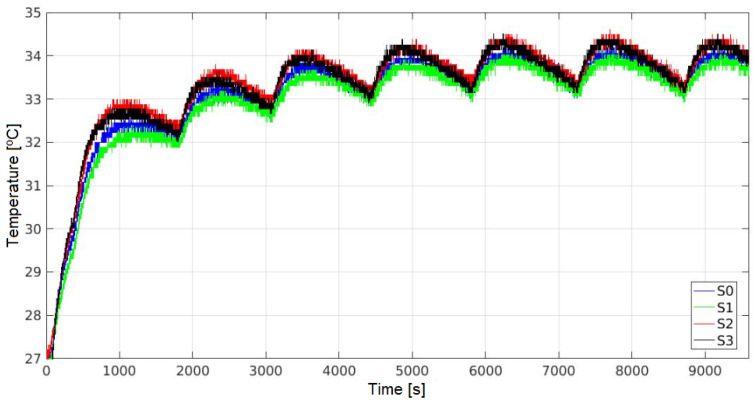
Values per sensor, with top shelf, including stabilization time.

**Figure 19 sensors-24-07671-f019:**
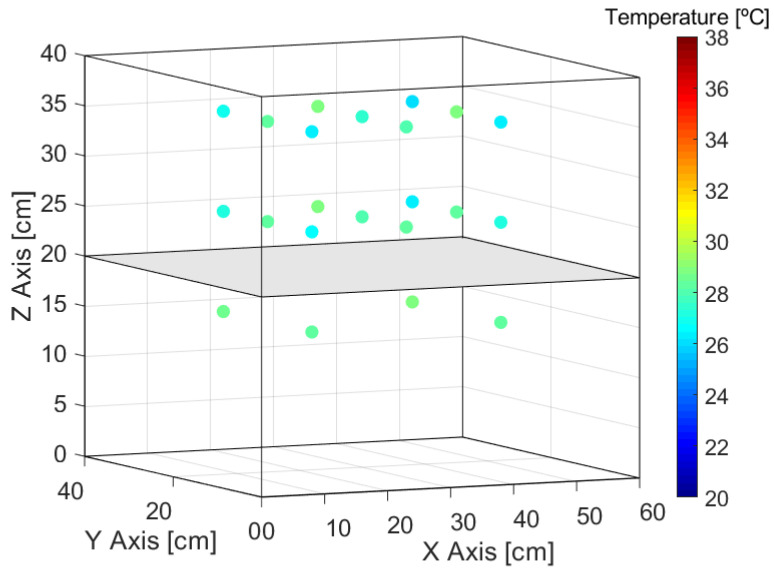
Measuring positions with shelf.

**Figure 20 sensors-24-07671-f020:**
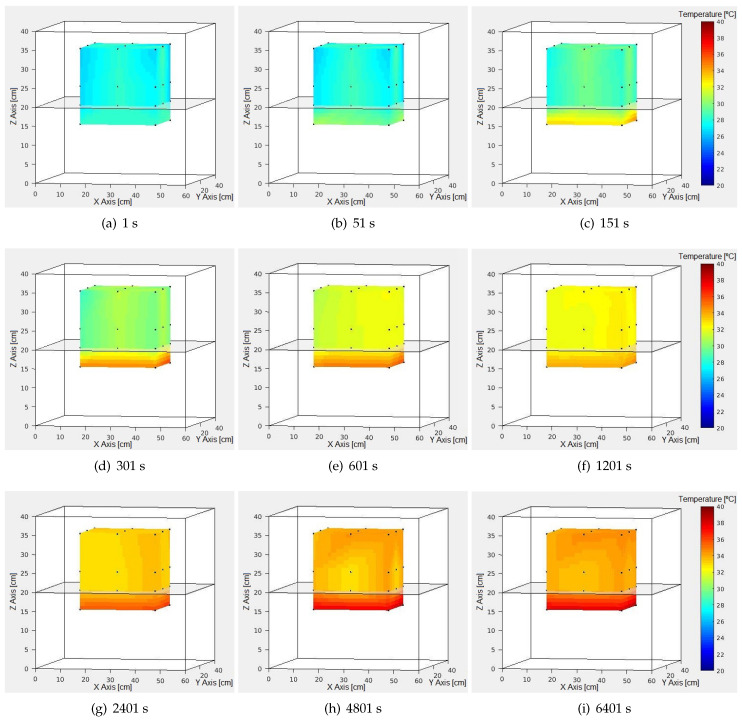
Internal thermal behavior of resized cube by time in seconds (s).

**Table 1 sensors-24-07671-t001:** List of components for device construction with four temperature sensors, configuration C.

Component	Amount
LM35 temperature sensor	4
2 kΩ resistor	4
47 nF multi-layer ceramic capacitor	4
Audiopipe HWY 836 USA standard MC-1 shielded cable	2.4 m
ARDUINO UNO board	1

Estimated cost values can be found in Appendix A.

**Table 2 sensors-24-07671-t002:** Representative section of results for a sensor with configurations B and C, respectively.

BC [°C]	CC [°C]
67.00	23.36
15.34	23.36
54.00	23.36
26.37	23.36
75.00	23.56
04.89	23.36
68.00	23.36
33.25	23.36
71.00	23.36
14.00	23.36

**Table 3 sensors-24-07671-t003:** Sensor correction equation values.

Sensor	a [°C]	b [1/°C]	$R^{2}$
**0**	0.79±0.06	0.050±0.002	0.99029
**1**	1.12±0.07	0.048±0.002	0.99199
**2**	1.2±0.1	0.043±0.003	0.98304
**3**	0.5±0.1	0.067±0.006	0.96602

**Table 4 sensors-24-07671-t004:** Average maximum temperatures (°C) after stabilization time.

Block	Row	$S_{0}$ [°C]	$S_{1}$ [°C]	$S_{2}$ [°C]	$S_{3}$ [°C]	Average $T_{\max}$ per Block [°C]
B1	1	34.6	34.2	34.2	34.4	34.25
	2	34.8	34.3	34.1	34.2	
	3	34.6	34.2	34.1	34.2	
	4	34.4	34.0	33.9	33.8	
B2	1	34.9	34.9	34.9	34.7	35.02
	2	35.1	34.8	34.8	34.8	
	3	35.6	35.1	34.9	35.2	
	4	35.3	35.1	34.9	35.3	
B3	1	35.0	35.3	35.7	35.2	35.21
	2	35.3	35.1	35.1	35.2	
	3	35.7	35.0	35.0	35.3	
	4	35.1	35.2	34.8	35.4	
B4	1	35.5	35.6	35.5	35.5	35.57
	2	35.9	35.5	35.5	35.6	
	3	35.7	35.4	35.4	35.7	
	4	35.7	35.5	35.5	35.6	
B5	1	35.2	34.9	34.8	35.0	35.53
	2	35.8	35.4	35.0	35.3	
	3	35.7	35.8	35.7	36.3	
	4	35.9	35.9	35.8	36.0	

**Table 5 sensors-24-07671-t005:** Averages of minimum ($T_{min}$) and maximum ($T_{MAX}$) temperatures as well as the difference (GAP) between them.

Temperature [°C]	Upon Shelf	Under Shelf
** $T_{min}$ **	33.1	34.8
** $T_{MAX}$ **	34.2	38.1
** GAP **	1.1	3.3

**Table 6 sensors-24-07671-t006:** Temperature results for sensor and thermocouple for each selected position.

Position	Temperature [°C]				
	$S_{0}$	$S_{1}$	$S_{2}$	$S_{3}$	Thermocouple
P1	33.3	-	-	-	33
P2	-	35.1	-	-	35
P3	-	-	35.2	-	35
P4	-	-	-	34.3	34
P5	32.3	31.9	32.0	31.9	32
P6	-	-	-	32.0	32
P7	34.1	-	-	-	34
P8	-	-	32.0	-	32
P9	-	32.4	-	-	32
P10	34.2	34.3	34.0	34.3	34

## Data Availability

Data are contained within the article.

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
