# Peer review of "Development of a Multiple Temperature Sensors Device for the Characterization, Control and Monitoring of Microbiological Incubators"

_sensors, 2024, doi:10.3390/s24237671_

Round 1
Reviewer 1 Report
Comments and Suggestions for Authors
In this paper, a methodology for assembling and calibrating a lower cost data logger capable of measuring temperature from different locations within the same system was proposed. Its application in laboratory systems is demonstrated through the characterization of the internal thermal behavior of a laboratory oven. The proposed device represents a potential work tool for laboratory systems with a proven response range of 20 to 45 ºC. This manuscript should be revised according to the following suggestions.
Question 1: The paper needs to supplement the shortcomings of the current temperature measurement system in the laboratory.
Question 2: The results of other temperature measuring devices of the same type need to be supplemented in order to obtain the advantages of this study.
Question 3: The paper needs to point out the innovation of the temperature measurement system in this study.
Question 4: The conclusion should be abbreviated.
Question 5: The principle of temperature detection of the temperature measurement system developed in this study needs to be given.
Author Response
Please see the attachment.
Response to Reviewer 1 Comments |
||
1. Summary |
|
|
“In this paper, a methodology for assembling and calibrating a lower cost data logger capable of measuring temperature from different locations within the same system was proposed. Its application in laboratory systems is demonstrated through the characterization of the internal thermal behavior of a laboratory oven. The proposed device represents a potential work tool for laboratory systems with a proven response range of 20 to 45 ºC. This manuscript should be revised according to the following suggestions.” ANSWER/REBUTTAL: The authors are grateful for taking the time to review this manuscript. All queries from the Reviewer #1 have been positively received and subsequently addressed as they enriched the manuscript. Please find the detailed responses below and the corresponding revisions added in the re-submitted files. |
||
2. Questions for General Evaluation |
Reviewer’s Evaluation |
Response and Revisions |
Does the introduction provide sufficient background and include all relevant references? |
Can be improved |
The Introduction section has been revised, and modified aiming at attaining the required improvement. Please refer to the many interventions applied to the original version of the Introduction section. |
Is the research design appropriate? |
Yes |
|
Are the methods adequately described? |
Can be improved |
Efforts have been invested, based on the Reviewers’ queries, in order to improve the description of the methodologies used in the study. |
Are the results clearly presented? |
Yes |
|
Are the conclusions supported by the results? |
Yes |
|
3. Point-by-point response to Comments and Suggestions for Authors |
||
Comments 1: The paper needs to supplement the shortcomings of the current temperature measurement system in the laboratory. |
||
Response 1: ANSWER/REBUTTAL: The authors are grateful for the valuable observation. In particular, the limitations of most current temperature measurement devices are due to the fact that they typically operate using a single sensor or probe for their determination. Thus, if they are used in large enough systems where temperature fluctuations that may affect the system, monitoring systems may be unnoticed, hence leading to undesirable consequences (microbiological incubators operating with unknown temperature inhomogeneities producing altered and even deficient growth of the microbiological agents of interest, and consequently compromising the results and reproducibility of the work that is being carried out; for instance). The query’s issue has been clarified and better addressed in the introduction section, also accounting for the third comment and revisions from another reviewer. Please refer to the introduction section in the revised version of the manuscript. |
||
Comments 2: The results of other temperature measuring devices of the same type need to be supplemented in order to obtain the advantages of this study. |
||
Response 2: ANSWER/REBUTTAL: The authors appreciate the very valuable suggestion, as it may allow for completeness. Comparison between the sensors of the developed device and mercury thermometers' readouts has been performed and reported in the “Device calibration” section. Furthermore, an extra subsection entitled “Proposed device evaluation against thermocouple” has been added in the “Materials and Methods” section aimed at boosting the cross-comparisons’ approach. The added subsection briefly depicts the cross-comparisons' procedure to assess the performance of the developed and already calibrated device against independent devices of the same type: a thermocouple. Moreover, an analogue subsection has been included in the “Results” section reporting the temperature measurements as obtained by both systems: the developed data logger and thermocouple. Corresponding advantages and disadvantages are briefly addressed. Please refer to the subsections referred above, in pages 8 and 16; respectively. |
||
Comments 3: The paper needs to point out the innovation of the temperature measurement system in this study. |
||
Response 3: ANSWER/REBUTTAL: The authors are grateful for pointing out this issue. According to first comment (item 1 of this list), authors have modified the introduction section to better and explicitly state the innovation of the proposed system, which mainly regards the development, implementation, and characterization of an instrument easy to recreate, economically low-cost, and capable of measuring the temperature inside the system from multiple points simultaneously, along with the development of a robust methodology for its use as monitor for the internal characterization of a laboratory stove. Please refer to the introduction section in the revised version of the manuscript. |
||
Comments 4: The conclusion should be abbreviated. |
||
Response 4: ANSWER/REBUTTAL: The authors appreciate the indication, which has been addressed by shortening the text for the conclusions also accounting for the second and sixth queries required by the reviewer #3. In this regard, direct remarks have been incorporated to state key features, such as the straightforward elaboration and low-cost of the device, along with the crucial contributions, practical applications, and potential future developments of the present study. Therefore, aiming at emphasizing the mentioned issues, the Conclusions section has been modified as follows: Original conclusion: “This work proposes a methodology for assembling and calibrating a lower cost data logger capable of measuring temperature from different locations within the same system. It was successfully built and calibrated. Furthermore, its application in laboratory systems is demonstrated through the characterization of the internal thermal behavior of a laboratory oven. The proposed device represents a potential work tool for laboratory systems, that can be extended to other research environments, with a proven response range of 20 to 45 ºC. In the absence of the oven's upper shelf, the stabilization time is 3000 seconds (s), while in the presence of the shelf the time extends to 5800 s. The temperature inside the oven tends to increase along the X axis, with a maximum gap of 1.5 ºC between side walls. Heat currents are suggested by the behaviors observed along the Z and Y axes, prompting a recommendation to resize the work area to a cube of 30 x 30 x cm3 concentric with the center of the oven, where the temperature variation is significantly reduced to 0.4 ºC. The controller sensor is calibrated to achieve an average maximum internal temperature of 37 ºC. Which is achieved with a maximum operating temperature configuration of 37 ºC, 2 ºC hysteresis and a correction of -1.4 ºC the temperature record of the controller sensor. With the upper shelf in the resized work area, the maximum temperature recorded under the shelf exceeds that above it. An average maximum temperature of 37 ºC is achieved by setting the maximum operating temperature at 37 ºC with 2 ºC hysteresis and applying a -2.0 ºC correction to the controller sensor's register. However, a maximum operating temperature setting of 37.8 ºC with 2 ºC hysteresis and a correction of -1.4 ºC is considered suitable.” Abbreviated and revised conclusion: “This work proposes a methodology for assembling and calibrating a low-cost data logger. Which is highly accessible to researchers and laboratories with limited resources due to the affordability and common availability of its components. Furthermore, the device is straightforward to construct and is capable of measuring temperature from different locations within the same system simultaneously, facilitating a three-dimensional characterization of temperature and its evolution over time. The proposed device was successfully built and calibrated. Additionally, its application in laboratory systems is demonstrated through a robust methodology of characterization of the internal thermal behavior of a laboratory oven. In the absence of the oven's upper shelf, the stabilization time is 3000 seconds (s), while in the presence of the shelf the time extends to 5800 s. The temperature inside the oven tends to increase along the X axis, and heat currents are suggested by the behaviors observed along the Z and Y axes, prompting a recommendation to resize the work area to a cube of 30 x 30 x 30 cm3 concentric with the center of the oven, where the temperature variation is significantly reduced to 0.4 ºC. In both cases, with and without the upper shelf, an average maximum temperature of 37 ºC can be achieved by setting the maximum operating temperature according to each case. The proposed device represents a potential work tool for laboratory systems, that can be extended to other research environments and with more locations, with a proven response range of 20 to 45 ºC that can also be extended if it’s needed. ” |
||
Comments 5: The principle of temperature detection of the temperature measurement system developed in this study needs to be given. |
||
Response 5: ANSWER/REBUTTAL: The authors appreciate the indication and have proceeded to explain the principle of temperature detection as follows: Original sentence: “The LM 35 sensors provide a temperature-proportional voltage output in ºC, with a linear reading factor increasing the value at the rate of 10 mV per ºC. They have a measurement range of -55 ºC to 150 ºC, with an accuracy of 0.5 ºC at room temperature (Tamb)” Corrected sentence: “The LM 35 sensors provide a temperature-proportional voltage output in ºC, with a linear reading factor increasing the value at the rate of 10 mV per ºC. Thus, meaning that if the output is, for example, 0 mV at 0 ºC, 250 mV at 25 ºC and 400 mV at 40 ºC. Temperature is assessed by sensors’ readouts due to the basic operation principle of diodes, as sketched in figure 2. It is well-known from semiconductor physics that the voltage across the diode increases at a known rate as the temperature increases, generating a signal directly proportional to the thermal variations [Grundmann]. Thus, the Shockley equation mathematically describes the relationship between the diode current (I) and the absolute temperature (T) according to the following expression: I = IS(e (qV/kT) -1) where Is is the reverse bias saturation current, q is the electron charge, V is the voltage across the diode and k is the Boltzmann's constant [Mansoor et al.]. It is worth mentioning that LM35 sensors have a measurement range of −55 ºC to 150 ºC, with an accuracy of 0.5 ºC at room temperature (Tamb)” References: ● Grundmann, M. The Physics of Semiconductors, An Introduction Including Nanophysics and Applications; Springer, 2021. https://doi.org/10.1007/978-3-030-51569-0. 451. ● Mansoor, M.; Haneef, I.; Akhtar, S.; De Luca, A.; Udrea, F. Silicon diode temperature sensors—A review of applications. Sensors and Actuators A: Physical 2015, 232, 63–74. https://doi.org/https://doi.org/10.1016/j.sna.2015.04.022. 453. Please, refer to page 3 line 73 and Figure 2 in the revised version of the manuscript. |
Reviewer 2 Report
Comments and Suggestions for Authors
Dear authors,
The work is about assembling and calibration of a lower cost data acquisition system for temperature measurement at various locations inside incubators.
The work is correct and clearly presented. The discussion and conclusions are interesting and pertinent.
However, I have a few questions:
-On page 11 line 213, it would be interesting for the reader to have the average temperature values per block . Why not including them in and additional column on Table 3 ?
-On Fig 12, all the sub figures are not necessary, they are rather qualitative … I would suggest to keep 9 sub figures corresponding to 9 times (as you did on Fig.15)
- On page 6 line 151, it would be interesting to provide the accuracy value of the LM35 sensor coming from the manufacturers?
and also I have a few suggestions for minor corrections:
-P. 3 line 76 : please suppress “ is used “ at the end of the sentence . One verb is enough.
-On table 2, please provide units for a, b and R squared.
-On page 8 line 188, replace “temperature” by “temperatures”
-On page 14, line 265, replace “compared the other” by “compared to the other”
Best regards
Author Response
Please see the attachment.
Response to Reviewer 2 Comments |
||
1. Summary |
|
|
“The work is about assembling and calibration of a lower cost data acquisition system for temperature measurement at various locations inside incubators. The work is correct and clearly presented. The discussion and conclusions are interesting and pertinent.” ANSWER/REBUTTAL: The authors are grateful for the positive reception and all suggestions and comments provided, allowing to improve the quality and presentation of the proposed work. Please find the detailed responses below and the corresponding revisions added in the re-submitted files. |
||
2. Questions for General Evaluation |
Reviewer’s Evaluation |
Response and Revisions |
Does the introduction provide sufficient background and include all relevant references? |
Yes |
|
Is the research design appropriate? |
Yes |
|
Are the methods adequately described? |
Yes |
|
Are the results clearly presented? |
Yes |
|
Are the conclusions supported by the results? |
Yes |
|
3. Point-by-point response to Comments and Suggestions for Authors |
||
Comments 1: On page 11 line 213, it would be interesting for the reader to have the average temperature values per block . Why not including them in and additional column on Table 3? |
||
Response 1: ANSWER/REBUTTAL: Thank you for pointing this out. The authors agree with this comment. Therefore, average temperature values per block have been added into a new column on Table (after revisions) 4 and the corresponding text was modified accordingly, as can be appreciated in page 13 line 261 in the revised version of the manuscript: “If the averages of the maximum temperatures per block are calculated” Has been replaced by “As can be seen in the averages from table 4” |
||
Comments 2: On Fig 12, all the sub figures are not necessary, they are rather qualitative … I would suggest to keep 9 sub figures corresponding to 9 times (as you did on Fig.15) |
||
Response 2: ANSWER/REBUTTAL: Agree. The authors have proceeded accordingly by selecting 9 sub figures corresponding to 9 readout times in Figure (after revisions) 17, taking into account the number of subfigures in Figure (after revisions) 20. Please, refer to Fig. 17 in the revised version of the manuscript. |
||
Comments 3: On page 6 line 151, it would be interesting to provide the accuracy value of the LM35 sensor coming from the manufacturers? Response 3: ANSWER/REBUTTAL: Thank you for pointing this out. The authors agree with this comment. Thereby and accordingly, added the accuracy value of the LM35 sensor informed by the manufacturers has been provided as follows: Original sentence: “In contrast, the CC configuration demonstrates remarkable stability, with a maximum variation of 0.20 ºC, which falls within the instrumental error margin, and an average recorded temperature of 23.38 ºC.” Corrected sentence: “In contrast, the CC configuration demonstrates remarkable stability, with a maximum variation of 0.20 ºC, which falls within the 0.5 ºC instrumental error margin, and an average recorded temperature of 23.38 ºC.” Please, refer to page 9 line 197 in the revised version of the manuscript. Comments 4: P. 3 line 76 : please suppress “ is used “ at the end of the sentence . One verb is enough Response 4: ANSWER/REBUTTAL: The authors appreciate the correction that has been addressed accordingly. Original sentence: “Each sensor uses a 60 cm of 4 mm diameter Audiopipe HWY 836 USA standard MC-1 shielded cable with two wires is used.” Modified sentence: “Each sensor uses a 60 cm of 4 mm diameter Audiopipe HWY 836 USA standard MC-1 shielded cable with two wires.” Please, refer to page 4 line 101 in the revised version of the manuscript. Comments 5: On table 2, please provide units for a, b and R squared. Response 5: ANSWER/REBUTTAL: Agree. The authors have proceeded accordingly providing corresponding units for a and b on table (after revisions) 3. However, the parameter R squared is a statistical (adimensional) index/measure with values ranging from 0 to 1. Please, refer to Table 3 in the revised version of the manuscript. Comments 6: On page 8 line 188, replace “temperature” by “temperatures” Response 6: ANSWER/REBUTTAL: The authors appreciate the indication. The redaction (mainly word's order) has been modified as follows: Original sentence: “Subsequently, T decreases until the W3230 sensor reaches 35 ºC (time B in Figure 10, at which point the resistors resume heat transmission, reaching a new peak temperature in the upper central area.” Modified sentence: “Subsequently, T decreases until the W3230 sensor reaches 35 ºC (time B in Figure 15, at which point the resistors resume heat transmission, reaching a new temperature peak in the upper central area.” Please, refer to page 11 line 233 in the revised version of the manuscript. Comments 7: On page 14, line 265, replace “compared the other” by “compared to the other” Response 7: ANSWER/REBUTTAL: The authors appreciate the indication, which has been addressed as follows: Original sentence: “This configuration presents less susceptibility to ground currents and greater stability compared the other configurations.” Modified sentence: “This configuration presents less susceptibility to ground currents and greater stability as compared to the other configurations.” |
||
After applying previous suggestions and corrections, this line remains on page 17, line 326. After applying suggestions and corrections, the number of tables, figures, pages and lines can be different in the revised version of the manuscript than the first version sent. |
Reviewer 3 Report
Comments and Suggestions for Authors
The paper focuses on the development of an instrumentation device for temperature measurement. From an electronics and implementation standpoint, the work is quite basic, similar to a typical undergraduate project. However, one positive aspect is that the calibration process is well-conceived and effectively documented. Additionally, the application for microbiological incubators is relevant and offers potential utility. In its current form, however, the work lacks a significant scientific contribution and may not yet be suitable for publication. I would recommend restructuring the paper to provide a more substantial contribution to the scientific community.
To enhance the relevance and originality of the study, I suggest framing it as a more comprehensive contribution under a title like "Development of an Open-Source, Low-Cost Multi-Sensor Device for Characterization, Control, and Monitoring of Microbiological Incubators." This would allow the authors to emphasize the value of open science by offering an open-source platform for users looking to develop or customize their own microbiological incubators. Additionally, it would highlight the low-cost, easily replicable nature of the device, making it accessible to a broader audience.
Additional Comments:
-
Introduction: I recommend including illustrative photos of microbiological incubators, showcasing potential applications and the importance of your project. This would help establish the context and relevance of the study.
-
Low-Cost and Ease of Construction: Emphasize that the device is low-cost and straightforward to construct, making it highly accessible to researchers and laboratories with limited resources. Highlighting the affordable components and simple assembly would increase the paper's appeal as a practical solution.
-
Instrument Configuration: The schematic in Figures 1 and 2 should emphasize Configuration C with a capacitor (for reducing power supply noise) and a resistor (for protecting the sensor and Arduino, stabilizing the signal, and filtering noise). A significant portion of the paper discusses basic instrumentation setups, ultimately concluding that Configuration C (referred to as CC) is optimal—a well-known recommendation in application notes. It would be more effective to proceed directly with Configuration C.
-
Design Requirements: I suggest the authors provide a list of design requirements, such as temperature range, autonomy, response time, power supply, and other critical factors for a biological incubator. This would help clarify the project scope and technical considerations.
-
System Block Diagrams: Block diagrams illustrating system integration (sensor connections, Arduino, computer interface, oven, etc.) would be highly beneficial. Additionally, installation recommendations for sensors would provide practical guidance for readers interested in replicating or modifying the setup.
-
Conclusion: The conclusion needs substantial improvement. It should summarize the key contributions, practical applications, and potential future developments of the device.
Author Response
Please see the attachment.
Response to Reviewer 3 Comments
- Summary
“The paper focuses on the development of an instrumentation device for temperature measurement. From an electronics and implementation standpoint, the work is quite basic, similar to a typical undergraduate project. However, one positive aspect is that the calibration process is well-conceived and effectively documented. Additionally, the application for microbiological incubators is relevant and offers potential utility. In its current form, however, the work lacks a significant scientific contribution and may not yet be suitable for publication. I would recommend restructuring the paper to provide a more substantial contribution to the scientific community.
To enhance the relevance and originality of the study, I suggest framing it as a more comprehensive contribution under a title like "Development of an Open-Source, Low-Cost Multi-Sensor Device for Characterization, Control, and Monitoring of Microbiological Incubators." This would allow the authors to emphasize the value of open science by offering an open-source platform for users looking to develop or customize their own microbiological incubators. Additionally, it would highlight the low-cost, easily replicable nature of the device, making it accessible to a broader audience.”
ANSWER/REBUTTAL:
Thank you very much for taking the time to review this manuscript. Please find below the detailed answers along with the corresponding revisions/corrections highlighted/in track changes in the re-submitted files.
2. Questions for General Evaluation |
Reviewer’s Evaluation |
Response and Revisions |
Does the introduction provide sufficient background and include all relevant references? |
Must be improved |
Significant efforts have been invested addressing all the Reviewers’ queries to improve the Introduction section. |
Is the research design appropriate? |
Must be improved |
Better descriptions are provided about the research design, thus aiming -and expecting- at attaining the required standards. |
Are the methods adequately described? |
Must be improved |
Substantial improvements have been achieved in terms of methodologies’ description as a consequence of addressing the Reviewers’ queries. |
Are the results clearly presented? |
Must be improved |
The Results section has been modified/enhanced in accordance with all the Reviewers’ queries. |
Are the conclusions supported by the results? |
Must be improved |
The Conclusions section has been significantly re-elaborated accounting for all the Reviewers’ queries. |
- Point-by-point response to Comments and Suggestions for Authors
Comments 1: Introduction: I recommend including illustrative photos of microbiological incubators, showcasing potential applications and the importance of your project. This would help establish the context and relevance of the study.
Response 1:
ANSWER/REBUTTAL: The authors appreciate the indication proceeding as follows:
- Example pictures/photographs of microbiological incubators have been added, with the temperature sensor position of each indicated, please refer to figure 1 in the revised version of the manuscript.
- Introduction has been strongly reformulated including potential applications and depicting the challenging context that has led to the current work, along with issues related to the second request highlighting the low cost and ease of construction of the device.
Please, refer to the introduction in the revised version of the manuscript.
Comments 2: Low-Cost and Ease of Construction: Emphasize that the device is low-cost and straightforward to construct, making it highly accessible to researchers and laboratories with limited resources. Highlighting the affordable components and simple assembly would increase the paper's appeal as a practical solution.
Response 2:
ANSWER/REBUTTAL: The authors appreciate the indication proceeding to emphasize these features in the revised/ corrected version of the manuscript, as follows:
- Introduction:
- Original sentence: “... for assembling and calibrating a data logger device, capable of simultaneously … ”
- Modified/corrected sentence: “... for assembling and calibrating a data logger device, simple to construct, low-cost, and capable of simultaneously …”
Please, refer to page 2 line 43 in the revised version of the manuscript.
- Materials and Methods:
- The following sentence was added along with its corresponding table: “In table 1 the list of necessary components to construct the device with configuration C, which includes the components of configurations A and B, is shown.”.
Please, refer to page 4 line 107 and table 1 in the revised version of the manuscript.
- Conclusion:
- Original sentence: “This work proposes a methodology for assembling and calibrating a lower cost data logger capable of measuring temperature from different locations within the same system.”
- Modified sentence: “This work proposes a methodology for assembling and calibrating a low-cost data logger. Which is highly accessible to researchers and laboratories with limited resources due to the affordability and common availability of its components. Furthermore, the device is straightforward to construct and is capable of measuring temperature from different locations within the same system simultaneously, facilitating a three-dimensional characterization of temperature and its evolution over time.”
Please, refer to the page 18 line 364 in the revised version of the manuscript.
Comments 3: Instrument Configuration: The schematic in Figures 1 and 2 should emphasize Configuration C with a capacitor (for reducing power supply noise) and a resistor (for protecting the sensor and Arduino, stabilizing the signal, and filtering noise). A significant portion of the paper discusses basic instrumentation setups, ultimately concluding that Configuration C (referred to as CC) is optimal—a well-known recommendation in application notes. It would be more effective to proceed directly with Configuration C.
Response 3:
ANSWER/REBUTTAL: The authors appreciate this suggestion. Nonetheless, it is worth mentioning that the present work is focused to a readers’ public that might not necessarily consist of experts on the different involved fields. Therefore, considering that many manufacturers/providers use to suggest configurations not consistent with the CC setup (please, refer to the following links), the authors would like to retain comparisons among CC and others configurations.
- https://www.alldatasheet.com/datasheet-pdf/view/1188696/TGS/LM35.html
- https://iopscience.iop.org/article/10.1088/1757-899X/163/1/012046/pdf
- https://polipapers.upv.es/index.php/JARTE/article/view/17392/15252
- https://www.researchgate.net/publication/303553647_Fire_Accident_Detection_and_Prevention_monitoring_System_using_Wireless_Sensor_Network_enabled_Android_Application
Despite the authors’ proposal for maintaining the comparisons with AA and BB configurations, authors are completely open to removing them completely according to the Reviewers/Editor suggestions.
Comments 4: Design Requirements: I suggest the authors provide a list of design requirements, such as temperature range, autonomy, response time, power supply, and other critical factors for a biological incubator. This would help clarify the project scope and technical considerations.
Response 4:
ANSWER/REBUTTAL: Authors really appreciate the possibility of clarifying this point. Thus, the following sentence has been added to the materials and methods section of the manuscript:
Added sentence: “The device to be developed is intended to be permanently operated according to typical microbiological incubators’ requirements, i.e. temperature ranging from 20 to 45 ºC; as well as being low-cost and simple to elaborate, along with independent temperature measurement by each sensor with a response time of at least 1 second. The device is intended to be fully supplied with voltages suitable for the ARDUINO UNO board without additional adaptations, making it suitable for extension and application in various systems.”
Please refer to page 2 line 57 in the revised version of the manuscript.
Comments 5: System Block Diagrams: Block diagrams illustrating system integration (sensor connections, Arduino, computer interface, oven, etc.) would be highly beneficial. Additionally, installation recommendations for sensors would provide practical guidance for readers interested in replicating or modifying the setup.
Response 5:
ANSWER/REBUTTAL: Authors appreciate the indication and the opportunity to better describe the whole system integration. Therefore, the following sentences have been added in page 7 line 166 along with the diagram in Figure 10, as a complement to Figure 9. Please refer to the revised version of the manuscript.
Added sentence: “Figure 10 represents the communication diagram for measurements of the whole system with the sensors in their respective positions according to the segmentation indicated in figure 9.”.
Comments 6: Conclusion: The conclusion needs substantial improvement. It should summarize the key contributions, practical applications, and potential future developments of the device.
Response 6:
ANSWER/REBUTTAL: The authors appreciate the indication that is consistently related to the first and second queries. Accordingly, the authors have modified the Conclusion section as follows:
- The low-cost and facility of construction of the device has been added and highlighted
- The key contributions such as the ability to measure temperature from multiple positions simultaneously that facilitates three-dimensional temperature characterization within the system, a practical application by its use in a laboratory oven for microbiological culture, along with potential future developments of the device had been added and exemplified.
Please, refer to the conclusion in the revised version of the manuscript.
Round 2
Reviewer 1 Report
Comments and Suggestions for Authors
It can be accepted.
Author Response
Response to Reviewer 1 Comments |
||
1. Summary |
|
|
“It can be accepted.” ANSWER/REBUTTAL: The authors are grateful for taking the time to review and accept this manuscript after revisions. |
||
2. Questions for General Evaluation |
Reviewer’s Evaluation |
Response and Revisions |
Does the introduction provide sufficient background and include all relevant references? |
Yes |
|
Is the research design appropriate? |
Yes |
|
Are the methods adequately described? |
Yes |
|
Are the results clearly presented? |
Yes |
|
Are the conclusions supported by the results? |
Yes |
|

Reviewer 3 Report
Comments and Suggestions for Authors
I would like to extend my congratulations to the authors on their work, which has shown significant improvement since the previous version. However, I would like to reiterate my concern regarding the inclusion of schematics A and B in Figure 3. These diagrams appear overly simplistic and might give the impression of basic or overly naïve instrumentation, which may detract from the overall quality of the manuscript. I suggest that the authors consider removing these schematics from the paper. Additionally, I recommend that the conclusion section be further refined to better emphasize the key findings and contributions of the study. Particular attention should be given to highlighting the significance of the low-cost, open-platform approach discussed in the paper. This would enhance the overall impact and clarity of the authors' conclusions. Lastly, I advise that the abstract be revisited to ensure alignment with the revised content of the manuscript. A clear and cohesive abstract will help to succinctly convey the scope, findings, and contributions of the research to readers. I look forward to seeing the final version of the paper.
Author Response
Please see the attachment
Response to Reviewer 3 Comments |
||
1. Summary |
|
|
“I would like to extend my congratulations to the authors on their work, which has shown significant improvement since the previous version. However, I would like to reiterate my concern regarding the inclusion of schematics A and B in Figure 3. These diagrams appear overly simplistic and might give the impression of basic or overly naïve instrumentation, which may detract from the overall quality of the manuscript. I suggest that the authors consider removing these schematics from the paper. Additionally, I recommend that the conclusion section be further refined to better emphasize the key findings and contributions of the study. Particular attention should be given to highlighting the significance of the low-cost, open-platform approach discussed in the paper. This would enhance the overall impact and clarity of the authors' conclusions. Lastly, I advise that the abstract be revisited to ensure alignment with the revised content of the manuscript. A clear and cohesive abstract will help to succinctly convey the scope, findings, and contributions of the research to readers. I look forward to seeing the final version of the paper.” ANSWER/REBUTTAL: The authors express their gratitude for the positive reception, which suggests the potential for acceptance following minor revisions. They also sincerely appreciate the valuable suggestions and comments provided, which have greatly contributed to enhancing the quality and presentation of the proposed work. Figure 3 has been modified, maintaining only configuration C, and the diagrams of configurations A and B have been moved to the supporting material. Please refer to Figure 3 in the revised version of the manuscript. As mentioned in the previous review, the authors consider their comparison important because the present work is focused on a readers’ public that might not necessarily consist of experts on the different involved fields and that many manufacturers use to suggest these types of simplistic configurations as shown in some examples of the following links:
Additionally, the Conclusions section has been modified to clarify and better emphasize the key findings and contributions of the study. Please refer to the Conclusions section in the revised version of the manuscript. Finally, the Abstract has been revised and completed in correspondence with the revised content of the manuscript. Please, refer to the abstract in the revised version of the manuscript. |
||
2. Questions for General Evaluation |
Reviewer’s Evaluation |
Response and Revisions |
Does the introduction provide sufficient background and include all relevant references? |
Yes |
|
Is the research design appropriate? |
Yes |
|
Are the methods adequately described? |
Yes |
|
Are the results clearly presented? |
Yes |
|
Are the conclusions supported by the results? |
Must be improved |
The Conclusions section has been modified/enhanced in accordance with the Reviewer queries |
